# Examination of the Complex Molecular Landscape in Obesity and Type 2 Diabetes

**DOI:** 10.3390/ijms25094781

**Published:** 2024-04-27

**Authors:** Uladzislau Vadadokhau, Imre Varga, Miklós Káplár, Miklós Emri, Éva Csősz

**Affiliations:** 1Proteomics Core Facility, Department of Biochemistry and Molecular Biology, Faculty of Medicine, University of Debrecen, 4032 Debrecen, Hungary; 2Doctoral School of Molecular Cellular and Immune Biology, University of Debrecen, 4032 Debrecen, Hungary; 3Department of IT Systems and Networks, Faculty of Informatics, University of Debrecen, 4028 Debrecen, Hungary; varga.imre@inf.unideb.hu; 4Department of Internal Medicine, Faculty of Medicine, University of Debrecen, 4032 Debrecen, Hungary; kaplar.miklos@med.unideb.hu; 5Department of Medical Imaging, Faculty of Medicine, University of Debrecen, 4032 Debrecen, Hungary; emri.miklos@med.unideb.hu

**Keywords:** type 2 diabetes, obesity, data integration, proteomics, metabolomics

## Abstract

The escalating prevalence of metabolic disorders, notably type 2 diabetes (T2D) and obesity, presents a critical global health challenge, necessitating deeper insights into their molecular underpinnings. Our study integrates proteomics and metabolomics analyses to delineate the complex molecular landscapes associated with T2D and obesity. Leveraging data from 130 subjects, including individuals with T2D and obesity as well as healthy controls, we elucidate distinct molecular signatures and identify novel biomarkers indicative of disease progression. Our comprehensive characterization of cardiometabolic proteins and serum metabolites unveils intricate networks of biomolecular interactions and highlights differential protein expression patterns between T2D and obesity cohorts. Pathway enrichment analyses reveal unique mechanisms underlying disease development and progression, while correlation analyses elucidate the interplay between proteomics, metabolomics, and clinical parameters. Furthermore, network analyses underscore the interconnectedness of cardiometabolic proteins and provide insights into their roles in disease pathogenesis. Our findings may help to refine diagnostic strategies and inform the development of personalized interventions, heralding a new era in precision medicine and healthcare innovation. Through the integration of multi-omics approaches and advanced analytics, our study offers a crucial framework for deciphering the intricate molecular underpinnings of metabolic disorders and paving the way for transformative therapeutic strategies.

## 1. Introduction

The rise in metabolic disorders, including type 2 diabetes (T2D) and obesity, presents a significant global health challenge, with profound implications for individual well-being and healthcare systems worldwide [1]. Both T2D and obesity are multifactorial conditions influenced by complex interplays between genetic predisposition, environmental factors, and lifestyle choices [2]. Understanding the underlying molecular mechanisms and identifying biomarkers associated with these conditions is crucial for early detection, targeted intervention, and personalized treatment strategies [3] as well as for prediction of complications (cardiovascular disease, diabetic nephropathy, and retinopathy).

Proteomics and metabolomics have emerged as powerful tools for unravelling the intricate molecular signatures characteristic of metabolic disorders [4]. By analyzing the comprehensive profiles of proteins and metabolites in biological samples, researchers can gain valuable insights into disease pathogenesis, identify diagnostic markers, and elucidate potential therapeutic targets [5]. Through correlation analyses integrating proteomics, metabolomics, and clinical parameters, researchers aim to uncover intricate networks of biomolecular interactions, identify key mediators of metabolic dysfunction, and delineate potential diagnostic and therapeutic avenues [6].

Proximity Extension Assay (PEA) technology utilizes antibody pairs labeled with oligonucleotide barcodes that hybridize when the antibodies recognize and bind the target proteins; hence, they are in a proximity. Hybridization is followed by extension and detection by quantitative polymerase chain reaction (qPCR) or next-generation sequencing (NGS). This enables avoiding cross-reactivity compared to classical immunoassays [7]. PEA has become a widely used method in proteomics research of cardiovascular and metabolic diseases. For instance, Vavruch et al. used PEA technology to find association between the level of 92 cardiovascular proteins and the amount of leptin to obtain better understanding of how leptin is linked to cardiovascular disease in patients with T2D [8]. In another study, Kozakova et al. assessed the connection between matrix-metalloproteinases, interleukins, lipids, and various cardiovascular parameters in both T2D and non-T2D cohorts [9].

In this study, we conducted an integrated analysis of serum proteomics and metabolomics data from a cohort comprising individuals with T2D and obesity as well as healthy controls. We aimed to delineate the distinct molecular landscapes associated with T2D and obesity, unravel novel potential biomarkers indicative of disease progression, and explore the intricate interplay between metabolic dysregulation and protein expression patterns.

Our investigation encompassed an in-depth assessment of cardiometabolic proteins using PEA technology [10], coupled with comprehensive characterization of serum metabolites, including amino acids and biogenic amines examined earlier by our group [11]. By using complex data evaluation methods, we could obtain insights regarding the molecular signatures underpinning metabolic disorders, and our study contributes to advancing our understanding of disease pathogenesis, refining diagnostic strategies, and informing the development of targeted interventions tailored to individual patient profiles. Moreover, our findings hold the promise of unveiling novel biomarkers and therapeutic targets that may ultimately transform the management and treatment of T2D and obesity, heralding a new era in precision medicine and personalized healthcare [12].

Through the integration of multi-omics approaches and advanced analytics, our study represents a critical step toward deciphering the complex molecular underpinnings of metabolic disorders and paving the way for innovative strategies to mitigate their escalating global burden.

## 2. Results

Cardiovascular complications are the most prevalent complications in patients with T2D, and its risk is also increased in obesity. Their signs may be hidden and appear suddenly, when the possibility for interventions is limited [13]. In our study, we aimed to examine the proteins related to cardiovascular metabolism to obtain more information on the cardiometabolic status of the patients recruited into the study with emphasis on the transition from obesity to T2D. The protein levels were correlated to clinical parameters and to the results of other tests relevant for cardiac health.

### 2.1. Differential Expression Analysis Reveals Distinct Protein Profiles in Obesity and Type 2 Diabetes

Relative protein abundances regarding 367 proteins having a role in cardiac fitness/metabolism were acquired using the Olink’s Explore Cardiometabolic panel in form of a service. After the completion of the PEA analyses, Olink provided comprehensive data tables containing normalized protein expression (NPX) values, obtained through intra- and inter-run normalization processes, as well as log2-scaling (Appendix A). These datasets served as the foundation for our subsequent analyses.

Differential expression analyses were conducted to elucidate the protein expression profiles associated with obesity and T2D. Results of the statistical tests were represented as volcano plots, utilizing the EnhancedVolcano R package (v 1.18.0) (Figure 1).

The comparison between control subjects and individuals with obesity revealed significant alterations in the expression levels of 17 proteins (*p* < 0.05) (Figure 1A) out of which 3 proteins were more and 14 proteins were less expressed. With high statistical significance, proteins FABP4, FCN2, and LEP showed 2-fold, 2.6-fold, and 4-fold increases in expression in the group of individuals with obesity, respectively. Moreover, expression of IGFBP1 was 2.5-fold higher in control subjects (*p* < 0.00003). The contrasts of protein expression in control subjects and those with T2D uncovered 24 proteins with statistically significant differences in expression (*p* < 0.05) (Figure 1B) with 4 less expressed and 20 more expressed. The IGFBP2 protein showed 1.8-fold increase in expression with high statistical significance in control subjects. In contrast, proteins ACY1, CES1, FABP4, FCN2, GSTA1, GUSB, and LEP were significantly lower with high statistical significance in subjects with T2D. Furthermore, comparison of protein expression patterns between individuals with obesity and those with T2D identified six proteins that exhibited differential expression (*p* < 0.05) (Figure 1C). Among them, LEP and LPL proteins showed 1.85-fold and almost 1.5-fold lower expression in individuals with T2D. On the other hand, IGFBP1 showed 2-fold higher expression in subjects with T2D. Among other more-expressed proteins in patients with diabetes are CES1, GDF15, and GUSB proteins, which were previously associated to T2D [14].

The examination of the level of individual proteins indicates association with different parameters (Appendix A). Some of the patients recruited into the T2D group were obese as well, so it was logical to examine the level of the individual proteins in patients with obesity and in patients with T2D with and without obesity. This analysis showed that some proteins, such as COL1A1, COL6A3, FABP4, LEP, LGALS1, and THBS4, are obesity-specific, and GDF15, LPL, MCFD2, REG1A, REG1B, and TCN2 are T2D-specific (Appendix A). We could also observe proteins that were differentially changed in both obesity and T2D compared to controls. These proteins were ADH4, CA5A, CNDP1, FCN2, MMP7, PLAT, THOP1, and VSTM2. Moreover, proteins such as CHI3L1, CTSD, GUSB, IGFBPL1, NRCAM, PAG1, PRCP, SELE, and USP8 were more abundant in the group of patients with T2D having high BMI. Overall, our results highlight distinct expression signatures of proteins from the Olink Cardiometabolic panel in the case of obesity and T2D.

### 2.2. Differentially Expressed Proteins Are Responsible for Various Functions

To reach more understanding on the role of the altered proteins, we conducted a pathway-enrichment analysis utilizing Pathway Browser from Olink Insight (https://insight.olink.com/ accessed on 1 March 2024 (Figure 2). Interestingly, enrichment of proteins in the extracellular matrix organization category is unique to control vs. T2D comparison (Figure 2a). Therein, protein COL1A1 was more expressed, while protein CTSD was less expressed. Secondly, pathways responsible for visual transduction were enriched in control vs. T2D and obesity vs. T2D but not in control vs. obesity contrast. Moreover, proteins responsible for transport of small molecules, namely chylomicron remodeling and assembly of active LPL and LIPC lipase complexes, were not enriched in control vs. obesity comparison (Figure 2c). Different patterns of enrichment are observed in the case of immune system and signal transduction category pathways in all three compared cases. In summary, differentially expressed proteins are involved in a number of pathways that might underline possible mechanisms of disease development or progression.

### 2.3. Olink Data Correlates with Metabolomics and Clinical Data

In our previous study [11], we performed measurement of the concentration of 23 amino acids and 10 biogenic amines followed by correlation analysis with clinical data. In this work, we complemented previous observations with proteomics data by conducting a correlation analysis between proteomics, metabolomics, and clinical data (Appendix A).

Our analysis revealed the presence of clusters, groups of variables where we observed either strong positive or strong negative correlation (Figure 3). In fact, such parameters as the level of histidine (His), the glomerular filtration rate (GFR), the level of taurine (Tau), the body mass index (BMI), and the level of high-density lipoprotein (HDL) clustered together and showed strong negative correlation with the level of majority of proteins in the group of control subjects. For example, the level of histidine was in a strong negative correlation with ACE2, ANG, CCL14, CCL27, CDH1, COL18A1, FAM3C, FAS, GDF15, HSPG2, IL18BP, LACTB2, LILRB2, MFAP5, NPDC1, SPON2, ST6GAL1, THBS4, and THPO. From those, GDF15, LILRB2, NPDC1, and THPO showed the strongest negative correlation (Figure 3, Control). Moreover, the His together with the GFR showed a cluster where the GFR showed the strongest negative correlation with CST3, CSTB, CD95, FABP2, PRSS2, and PTGDS, while His was strongly negatively correlated with LTBP2, SCARF1, and TFF3. Besides the region with negative correlation, there is a region with clusters of strong positive correlation. The parameters that showed the strong positive correlation with the level of the majority of cardiometabolic proteins are the levels of aspartate (Asp), citrulline (Cit), glutamate (Glu), ethanolamine (Eth), and age. These parameters cluster together with proteins ABCA7, ABCC2, ACAN, ACY1, ACTA2, ADGRE5, ADH4, ALCAM, APOM, AZU1, CCL15, CD2AP, CD55, CD59, CDH6, CD95, CNTN1, COL6A3, CST3, CSTB, FABP2, GP1BA, GSTA1, IGFBP6, IL19, ITGB2, LEPR, LTBP2, NADK, OLR1, PCOLCE, PRSS2, PTGDS, REN, RNASET2, S100A11, SCARF1, SDC4, SORT1, SPP1, TFF3, THPO, and TIMD4. Of note, the level of proteins ACTA2 and TFF3 has strong positive correlation with all abovementioned parameters (Figure 3, Control). Other parameters that showed strong positive correlation with the level of proteins are the levels of tyrosine (Tyr), alanine (Ala), ornithine (Orn), leucine (Leu), isoleucine (Ile), phenylalanine (Phe), glycated hemoglobin (HbA1C), triglyceride, C-peptide, insulin, low-density lipoprotein (LDL), and cholesterol (Figure 3, Control).

Correlation analysis of the data obtained from the obesity group revealed four major clusters (Figure 3, Obesity). Interestingly, two sets of proteins can be defined. Namely, ACTA2, ALCAM, AMY2A, AMY2B, ANGPTL3, APOM, APLP1, BPIFB1, CCL14, CCL27, CCDC80, CD46, CD55, CHL1, CLUL1, CNTN1, COL18A1, CRTAC1, CXXC4, DCN, EFEMP1, ENPP2, FABP4, GPNMB, GH1, GHRL, ICAM1, IGFBP1, IGFBP2, IGSF8, ITGB1, ITGB2, ITIH3, KIT, KITLG, LEPR, LGALS1, LEP, LPL, LTBP2, MCAM, NPTXR, NPDC1, NOTCH3, NTproBNP, NPPB, PCOLCE, PAM, PLA2G1B, PLA2G2A, PLTP, PTN, RARRES2, REG1A, REG1B, REG3A, ROR1, SCARF1, SERPINA11, SPARCL1, TFF3, TCN2, THPO, TIMD4, TINAGL1, and XG were positively correlated with ApoA, HDL, and Gly, while negatively correlated with the level of Leu, Ile, triglyceride, waist–hip ratio (WHR), neck size, abdomen size, insulin level, homeostatic model assessment of insulin-resistance (HOMA), and the level of C-peptide. Conversely, proteins ACE2, ACY1, ADH4, AGXT, AK1, CA5A, CCL16, CEP43, CES1, CTSH, EIF4EBP1, FBP1, GSTA1, GPR37, GUSB, LDLR, LACTB2, MB, MSTN, PAG1, QDPR, SELE, SDC1, SERPINE1, SSC4D, SSC5D, STK11, THOP1, TYMP, UMOD, USP8 were positively correlated with the levels of Tyr, Glu, Phe, Leu, Ile, triglyceride, WHR, neck size, abdomen size, insulin, HOMA, and C-peptide, while negatively correlated with ApoA, HDL, and Gly.

Correlation analysis in the T2D group (Figure 3, Diabetes) revealed that the levels of Tau, Asp, Glu, and Ala showed a strong positive correlation with CCL27, CD2AP, CTSD, LACTB2, MARCO, NADK, PAG1, PDGFA, PLXNB3, PPP1R2, SCARF1, SDC4, SERPINE1, SORT1, SOST, and ST6GAL1 proteins. More clusters with strong positive correlations that are of interest include one with Eth and Orn where they correlated with ACAN, ADA2, ADGRE5, ANGPTL3, CD59, COL1A1, CTSL, DCTPP1, DCN, ENTPD6, GGH, ICAM1, LTBP2, NPTXR, PLA2G2A, PTN, SPP1, THPO, TIMP1, and TNFSF13B and another with a correlation between size of abdomen, waist, and BMI and CCL18, CCDC80, COL6A3, CSTB, CTSZ, ENPP2, GPNMB, IGSF8, LEP, LGALS1, PAM, PLXNB2, SEMA3F, TFRC, and TFP1. In contrast, CRP and the level of His showed a strong negative correlation with ACP5, ADGRG2, AOC3, ANG, ANPEP, AZU1, CCL14, CEBPB, CD69, CEP43, CHL1, CNST, COMP, CPB1, CPA1, CNTN1, DOK2, EGFR, GP1BA, GYS1, ITGB1, LCN2, LILRA5, MCAM, MCFD2, NCAM1, NPPB, NRP1, NtproBNP, OLR1, PRSS2, RARRES2, RNASE3, SNAP23, SPARCL1, STK11, TFRC, and USP8, respectively. Additionally, similarly to the clusters observed in the control group, parameters such as His, GFR, Tau, and BMI showed negative correlation with the majority of proteins from the cardiometabolic panel.

### 2.4. Cardiometabolic Proteins Interact with Each Other

To understand the holistic picture of cardiometabolic proteins’ role in the linkage of T2D with obesity, we generated protein networks for the cardiometabolic proteins (Figure 4).

The network of proteins shows a similar picture in both conditions as the examined proteins are the same. However, in order to highlight the differences between the diseased and the control groups, we calculated the quotient of the mean NPX values between the diseased population and control group in the case of each protein. Quotients around 1.0 mean negligible differences between the pathological and control groups. Quotients below 1.0 demonstrate decreased values, while quotients above 1.0 illustrate higher NPX values than in the control group.

The interaction network of proteins between control and obese or control and T2D groups, respectively, contains 15 clusters in both cases. Among the 15 clusters, there is a large one covering almost 75% of the nodes.

The network of cardiometabolic proteins in obesity showed that the strongly higher-expressed LEP in the obese group interacts with the less-expressed GHRL and more-expressed GH1 (Figure 4A). Interestingly, from the STRING database we found that LEP and GH1 have physical interaction with a high confidence score (0.9). In contrast, proteins HYAL1, GUSB, and CES1 created a separate interaction cluster (Figure 4). Considering the network of proteins between control and obese groups, GUSB and CES1 are slightly more expressed in individuals with obesity (Figure 4A), while significantly higher expression of both can be observed in T2D (Figure 4B).

Regarding the proteins with the highest connectivity, DCN, EGFR, ITGB1, ITGB2, and SDC1 are among the top 10 hub proteins. Additionally, some of the differentially expressed proteins have central locations in the network. SDC4 and IL6 were in the top 10 of the most-connected proteins.

Thus, the graph describes a quite complex biochemical system, which cannot be understood as a set of protein-pair interactions.

## 3. Discussion

The identification of new biomarkers is crucial for the diagnosis, prevention, and treatment of diseases. In the case of metabolic disorders, biomarkers can serve as valuable tools for early detection, risk stratification, and prediction of complications. Therefore, new confident biomarkers can facilitate improvement in patient outcomes, reduce disease burden, and advance the field of precision medicine.

The integration of proteomics and metabolomics data in our study provides possibility for a comprehensive examination of the molecular landscapes associated with T2D and obesity. Our findings underscore distinct molecular signatures and unveil novel potential biomarkers indicative of disease progression in these metabolic disorders. By leveraging advanced analytics and pathway enrichment analyses, we elucidate the functional implications of differential protein expression patterns and uncover unique mechanisms underlying disease development and progression. Our results align with previous studies utilizing PEA technology, which has been instrumental in uncovering associations between cardiovascular proteins and metabolic diseases [8,9].

The differential expression analysis highlights significant alterations in protein expression profiles between control subjects and individuals with T2D or obesity, revealing distinctive patterns reflective of disease pathology. We observed significant upregulation of proteins like COL1A1, COL6A3, FABP4, and LEP in individuals with obesity, indicating potential roles in adipose tissue dysfunction and metabolic dysregulation [15]. Conversely, subjects with T2D showed significant downregulation of LEP and upregulation of proteins such as CES1 and GSTA1, suggesting alterations in detoxification pathways and lipid metabolism [16]. The comparison between obesity and T2D cohorts revealed distinct expression patterns, further emphasizing the heterogeneity of metabolic disorders and the need for personalized approaches in diagnosis and treatment.

Pathway enrichment analyses further elucidate the involvement of proteins in crucial pathways related to extracellular matrix organization, visual transduction, and small molecule transport, underscoring the multifaceted nature of disease pathogenesis [17]. These pathways offer insights into potential therapeutic targets and avenues for intervention. The dysregulation of extracellular matrix proteins like COL1A1 and CTSD suggests alterations in tissue remodeling processes and may contribute to the development of metabolic complications, such as insulin resistance and cardiovascular diseases [18].

Correlation analyses between proteomics, metabolomics, and clinical parameters unveil intricate networks of biomolecular interactions and identify key mediators of metabolic dysfunction, offering valuable insights into disease mechanisms and potential therapeutic targets [19]. Strong correlations between His levels and several cardiometabolic proteins indicate potential regulatory roles of His in metabolic homeostasis and inflammation [20]. Conversely, positive correlations between amino acids like Asp and Glu with cardiometabolic proteins highlight the interplay between amino acid metabolism and cellular signaling pathways in metabolic disorders [21].

The protein–protein interaction networks provide a holistic view of cardiometabolic proteins’ roles in linking T2D with obesity, highlighting key interactions and potential avenues for further investigation [22]. The interactions between GH1 and LEP underscore the intricate crosstalk between adipose tissue and endocrine signaling pathways in metabolic regulation [23]. Conversely, the interactions between CES1, GUSB, and HYAL1 suggest coordinated regulation of metabolic processes and detoxification pathways in response to metabolic stress [24].

Monitoring the level of some proteins, such as the increase of CTSD, GDF15, GUSB, MCFD2, REG1A, REG1B, and TCN2 and concomitantly the decrease of FABP4, LEP, and LPL may help in identifying the group of patients with obesity with higher risk for the appearance of T2D allowing for more precise therapeutic interventions. Of course, further validation is required to prove the utility of the abovementioned proteins as potential prognostic markers for T2D in patients with obesity.

Besides recapitulating some previously discovered associations, the results of our study further highlight the importance of application of multi-omics approaches to show the tightly interconnected network of proteins and metabolites to uncover new biomarkers. These interrelations along with the heterogeneity observed at the group level emphasize the importance of further patient stratification aiming for personalized interventions and heralding a new era in precision medicine and healthcare innovation. However, our study possesses some limitations. For example, the number of recruited subjects is good for conducting an exploratory, cross-sectional study, but it does not allow us to generalize our findings. Therefore, future studies with more recruited patients are needed. It should be emphasized that biomarker studies heavily rely on validation experiments. Thus, an appropriate validation strategy with broader and diverse cohorts should be outlined in future research.

In multi-omics approaches, the complex nature of the metabolic diseases can be depicted with the integration of as many data layers as possible. Besides proteomics, metabolomics, genomics, and transcriptomics data are required to better understand the complex interrelations among the metabolic pathways. For instance, our future research aims to recruit more layers of data such as phosphoproteomics, complex metabolomics, and lipidomics, along with genomics and epigenomics. This approach could allow the improvement of data-driven discoveries and facilitate precision medicine for metabolic diseases; however, the complexity of the data necessitates the application of system biology and complex bioinformatics workflows while the utilization of artificial intelligence is inevitable. These types of multi-omics studies require interdisciplinary collaborations involving researchers with different backgrounds, such as clinicians, computational biologists, statisticians, and bioinformaticians. This interdisciplinary collaboration could provide a blueprint for translating research findings to bedside applications, reinforcing the study’s impact and applicability.

Through the integration of multi-omics approaches and advanced analytics, our study offers a crucial framework for deciphering the intricate molecular underpinnings of metabolic disorders and guiding transformative therapeutic strategies.

## 4. Materials and Methods

### 4.1. Study Subjects and Sample Collection

In total, samples from 130 subjects were collected, including 53 patients with T2D, 45 individuals with obesity, and 32 healthy volunteers. Sample collection was approved by the Ethics Committee of the University of Debrecen, and all participants provided written informed consent. The T2D group’s mean age was 50.7 years, with a “male-to-female” ratio of 1.4:1; the obese group’s mean age was 51.9 years, with a “male-to-female” ratio of 0.8:1; and the healthy group’s mean age was 49.3 years, with a “male-to-female” ratio of 1.5:1. Table 1 represents clinical characteristics of study groups, while Appendix A contains clinical characteristics per every subject in the study.

T2D was diagnosed with an oral glucose tolerance test (OGTT). The subjects were included in group of patients with obesity if BMI was ≥30 and T2D was not diagnosed.

Fasting blood samples were collected from all participants in native tubes and centrifuged to extract the serum. Sera were aliquoted and stored at −80 °C until they were processed.

### 4.2. Analysis of Cardiometabolic Proteins with Proximity Extension Assay

PEA of 369 proteins from the Olink Explore Cardiometabolic panel was performed by the analysis service of Olink Proteomics (Uppsala, Sweden). In short, 1 μL of plasma sample from each donor was used by Olink to provide the normalized NPX values for the 367 proteins from the panel. Raw values were converted to normalized protein expression units by intra- and inter-run normalization and log_2_-scaling [7]. Olink technology is described in detail on their website: https://www.olink.com/. The abbreviated names of the proteins included in the panel with their respective full names are provided as a Appendix A.

### 4.3. Data Analysis

Results from PEA were filtered, explored, and cleaned utilizing R (v 4.3.1) statistical programming language with in-house developed scripts. Cleaning included removal of observations with warnings or failures in quality checks assessed by Olink with incubation and detection controls [7]. Olink applies a very strict quality control system. An observation was excluded from the Olink results table if the assay warning or quality-check warning had the value “EXCLUDED”. In the case that these parameters were “PASS” or “WARN”, the observation was retained.

#### 4.3.1. Pathway Enrichment Analysis

Pathway enrichment analysis was performed in Olink Insight’s Pathway Browser (PB) powered by Reactome. Lists with differentially expressed proteins were loaded into PB, and enriched pathways were visualized.

#### 4.3.2. Statistical Analysis

A Kruskal–Wallis non-parametric test with a Benjamini–Hochberg [25] correction was calculated for every protein in the Olink data. Proteins that passed the adjusted *p*-value threshold of 0.05 were subjected to post hoc Dunn’s test with Benjamini–Hochberg adjustment for multiple comparison. Log_2_ fold-changes of proteins were calculated by subtracting mean expression values in compared groups. Finally, volcano plots were generated utilizing the EnhancedVolcano package in R by plotting log_2_ of fold-change versus −log10 of *p*-value.

Group differences between the groups of patients with T2D normal BMI, high BMI patients with obesity, and controls were calculated using the Kruskal–Wallis test and pairwise Wilcoxon tests.

#### 4.3.3. Correlation Analysis

The measurements of amino acids and biogenic amines in serum as well as clinical data were taken from our previous study [11]. Spearman’s correlation coefficient was calculated between variables utilizing R (v 4.3.1) statistical programming language with in-house developed scripts. *p*-value was adjusted with the false discovery rate (FDR) method [25].

#### 4.3.4. Network Analysis

Mean normalized protein expression (NPX) values provided by Olink for each protein were exponentially transformed (the base was 2).

Information on interacting proteins was obtained from String Database considering only interactions with the highest confidence (0.9) and exported to a text file. In order to create, analyze, and represent networks, the text files were imported to the Cytoscape 3.10.1 (Institute for Systems Biology, Seattle, WA, USA) open-source bioinformatics software platform. The positions of the nodes during the visualization were determined by the Perfuse Force Directed Layout algorithm. Using this software, the size and color of nodes and links could be configured according to the imported data.

## Figures and Tables

**Figure 1 ijms-25-04781-f001:**
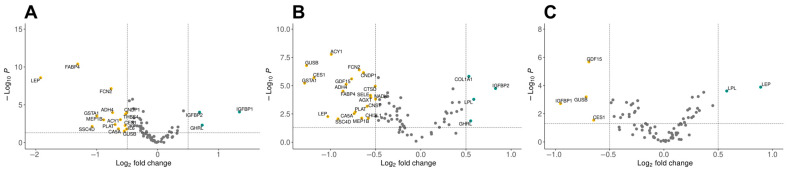
Differentially expressed proteins visualized as volcano plots where the x-axis represents log2 of the fold-change of protein expression, while the y-axis represents −log10 of the *p*-value after statistical analysis. (**A**) Differentially expressed proteins between control subjects and subjects with obesity; (**B**) Differentially expressed proteins between control subjects and subjects with type 2 diabetes (T2D); (**C**) Differentially expressed proteins between subjects with obesity and subjects with T2D.

**Figure 2 ijms-25-04781-f002:**
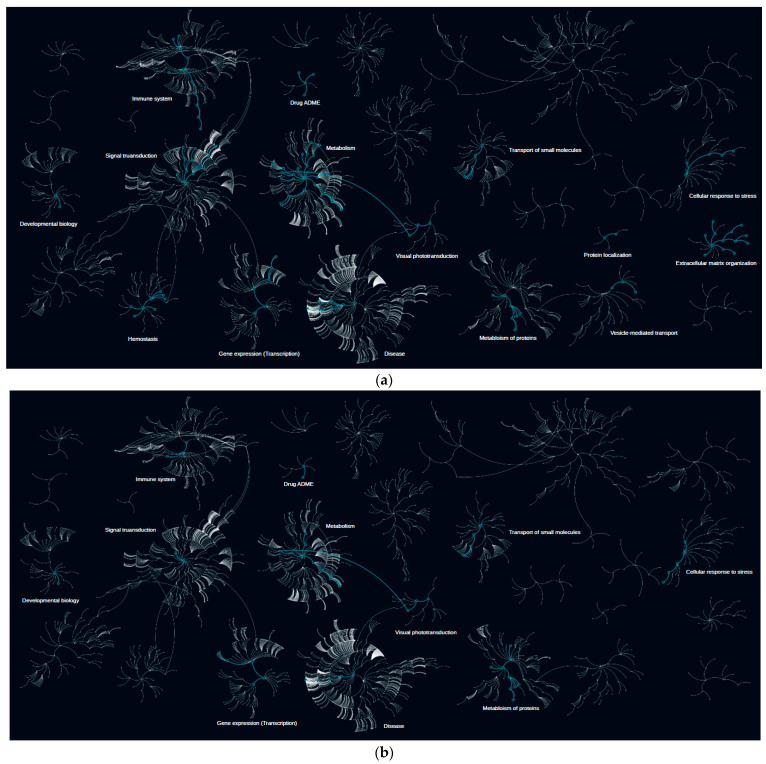
Pathway-enrichment analysis in the case of differentially expressed proteins. (**a**) Pathways enriched with proteins altered between control subjects and subjects with type 2 diabetes (T2D); (**b**) Pathways enriched with proteins altered between control subjects and subjects with obesity; (**c**) Pathways enriched with proteins altered between subjects with obesity and subjects with T2D. The higher resolution images of the networks are presented in Appendix A.

**Figure 3 ijms-25-04781-f003:**
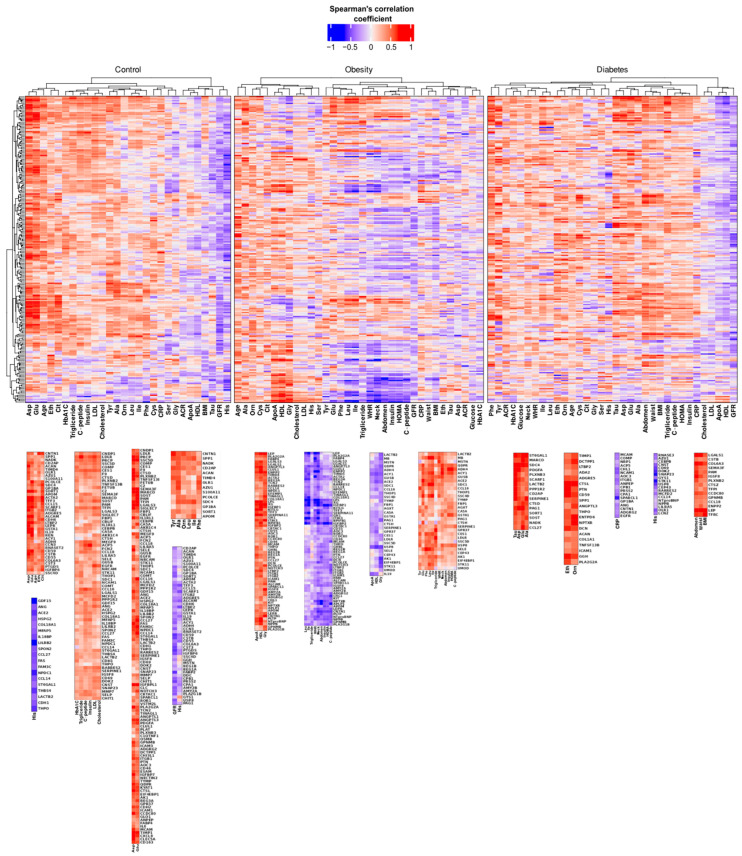
Correlation analysis. Spearman’s correlation coefficients were represented as a heatmap, followed by row and column clustering. Rows represent proteins, while columns represent either a clinical parameter, an amino acid, or a biogenic amine. Several clusters were highlighted under the heatmap.

**Figure 4 ijms-25-04781-f004:**
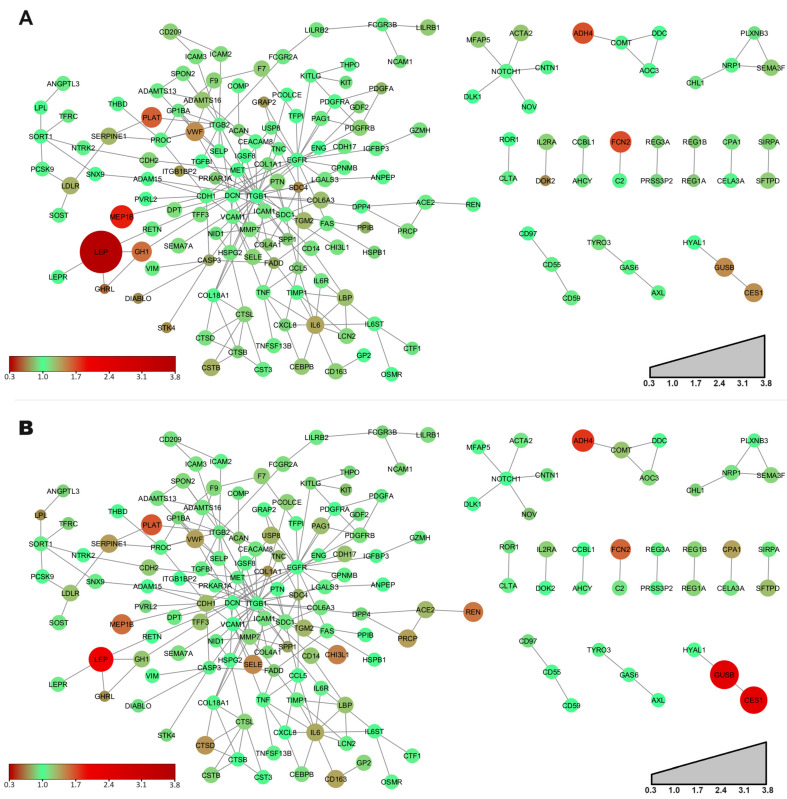
Network analysis of differentially expressed proteins. (**A**) Interaction network of proteins in comparison of protein expression between controls and subjects with obesity; (**B**) Interaction network of proteins in comparison of protein expression between controls and patients with type 2 diabetes (T2D). The size of the nodes is proportional to the obese/control or T2D/control NPX quotients. The green color indicates quotients close to 1, while the change of color toward dark brown and red indicates the deviation from the control group.

**Table 1 ijms-25-04781-t001:** Characteristics of the studied groups.

Parameter ^1^	Control	Obesity	Type 2 Diabetes
	(N = 32)	(N = 44)	(N = 54)
Age, years	49.25 ± 10.13	51.93 ± 9.61	50.72 ± 7.94
Sex (number)	Male (19)	Male (20)	Male (32)
	Female (13)	Female (24)	Female (22)
Body mass index (BMI)	25.33 ± 2.08	37.8 ± 5.82	33.78 ± 5.88

^1^ Age and BMI are indicated as mean ± SD.

## Data Availability

Data is contained within the article or in the Appendix A.

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
