# Peer review of "Examination of the Complex Molecular Landscape in Obesity and Type 2 Diabetes"

_ijms, 2024, doi:10.3390/ijms25094781_

Round 1

Reviewer 1 Report

Comments and Suggestions for Authors

The manuscript entitled “Examination of the complex molecular landscape in obesity and type 2 diabetes” and submitted in the International Journal of Molecular Sciences is devoted to the identifying of key proteins and molecular pathways underlaying obesity and type 2 diabetes using high-throughput methods followed by appropriate bioinformatical analysis.  The manuscript is well-written, the obtained results are very urgent for understanding of molecular basis of obesity and type 2 diabetes pathogenesis, but there are some issues that must be solved before publication.

1. Characteristic of studied groups must be presented in the manuscript. Authors must include the table with clinical characteristic of patients included in the presented research and show the differences by each parameter between groups;

2. How the obesity and type 2 diabetes were diagnosed? Which parameters were used to include patients in the studied groups? Which excluding criteria was applied? Authors must add this information in the Material and Methods section;

3. It is not clear from the methods description whether the authors used public availiable proteomic profiling databases or whether they performed this analysis themselves. The methods description ends with blood sampling from patients, and then bioinformatics analysis is described. Did the authors perform protein isolation, sample preparation for proteomic profiling, did they perform proteomic profiling and by what method? The authors should clarify what happened between blood collection from patients and data analysis;

4. The 4.3. Data analysis subsection is too short. Authors must clarify cut-off criteria that were used for data analysis;

5. If the authors themselves performed transcriptomic profiling, the data set must be uploaded to special depositories (e.g. PRIDE) and a link to it must be provided in the Data Availability Statement section.

Comments on the Quality of English Language

Minor editing of English language required.

Author Response

Dear Reviewer,

Thank you for contributing to our manuscript with your valuable comments. Please find the answers to them below.

Comment 1: Characteristic of studied groups must be presented in the manuscript. Authors must include the table with clinical characteristic of patients included in the presented research and show the differences by each parameter between groups.

Answer: It indeed increases the quality of the manuscript by adding the table with subjects’ parameters. Thank you for pointing to this. The table was created and added to the manuscript as Table 1. The table contains the number of patients recruited in each group, their sex, age (years), and body mass index. The mean value was calculated in each case along with standard deviation. Additionally, Supplementary Table S3 representing the above mentioned characteristics per every subject of the study (anonymity of patients is ensured) was created.

Comment 2: How the obesity and type 2 diabetes were diagnosed? Which parameters were used to include patients in the studied groups? Which excluding criteria was applied? Authors must add this information in the Material and Methods section.

Answer: Type 2 diabetes was diagnosed with oral glucose tolerance test (OGTT). The subjects were included in group of patients with obesity if BMI was ≥30 and T2D was not diagnosed. This information was added to the 4.1. Study subjects and Sample Collection section of Materials and Methods.

Comment 3: It is not clear from the methods description whether the authors used public available proteomics profiling databases or whether they performed this analysis themselves. The methods description ends with blood sampling from patients, and then bioinformatics analysis is described. Did the authors perform protein isolation, sample preparation for proteomic profiling, did they perform proteomics profiling and by what method? The authors should clarify what happened between blood collection from patients and data analysis.

Answer: The sample collection was followed by protein extension assay (PEA). The assay was ordered from Olink Proteomics company (Uppsala, Sweden) as a service. All sample were sent to the point of the assay in appropriate conditions to ensure the high quality of the results. The section 4.2 in the Methods part was renamed and updated to ensure the clarity of the experiment workflow.

Comment 4: The 4.3. Data analysis subsection is too short. Authors must clarify cut-off criteria that were used for data analysis.

Answer: Thank you very much for the comment. The 4.3. Data Analysis section was updated. More information on data analysis and cut-off values was added with the text “Results from PEA were filtered, explored, and cleaned utilising R (v 4.3.1) statistical programming language with in-house developed scripts. Cleaning included removal of observations with warnings or failures in quality checks assessed by Olink with incubation and detection controls [25]. The observation was excluded from the Olink results table if the assay warning or quality check warning had the value “EXCLUDED”. In case these parameters were “PASS” or “WARN”, the observation was retained.

Comment 5: If the authors themselves performed transcriptomics profiling, the data set must be uploaded to special depositories (e.g. PRIDE) and a link to it must be provided in the Data Availability Statement section.

Answer: The transcriptomics profiling was not performed by authors as well as no publicly available transcriptomics data was used. However, proteomics data was produced by authors, and the table containing the relative quantity values in the case of each examined protein is provided as a supplementary table (Table S1). The metabolomics data was taken from the previous publication from our lab (http://dx.doi.org/10.3390/ijms23094534).

Reviewer 2 Report

Comments and Suggestions for Authors

Dear Editor,

I am extremely grateful for the opportunity to review this valuable article.

The escalating prevalence of metabolic disorders, notably type 2 diabetes  and obesity, presents a critical global health challenge, necessitating deeper insights into their molecular underpinnings. The study integrates proteomics and metabolomics analyses to delineate the complex molecular landscapes.

My suggestion for authors:

Try avoid abreviations in abstract

Article looks like there are not enough informations in discusion section, not sufficient information in results seciotn and there are no conclusion.

Author Response

Dear Reviewer,

Thank you for contributing to our manuscript with your valuable comments. Please find the answers to them below.

Comment 1: Try avoid abbreviations in abstract.

Answer: Thank you for the comment. All abbreviations were removed from the abstract.

Comment 2: Article looks like there are not enough information in discussion section, not sufficient information in results section and there are no conclusion.

Answer: We are very sorry if the reviewer had the feeling that not enough information were provided in the mentioned sections. In the Results part we tried to give a detailed description of the acquired data and the results of the statistical, pathway and network analyses. At the same time, we tried to keep a good balance of data presentation not to overwhelm the reader with unnecessary details, so we tried to show only those the information in the main text which are necessary to understand the logic of our experimental design and the most important findings. All the other additional information were provided as supplementary material, either in form of table or figures.

In the Discussion part, based also on the comments of the other reviewers, we made substantial changes, hope it is adequate in the current form.

Regarding the Conclusion, as far as it is not a mandatory part of the manuscript, we were puzzling to either include it or not as a separate section. As far as we have more than one conclusion, the section would be a bit longer than usual, so we decided to implement it into the Discussion part.

We leave this question to the discretion of the Reviewer and Editor, and based on their comments we can make a separate Conclusion section restructuring the Discussion part as well, if it is needed.

Reviewer 3 Report

Comments and Suggestions for Authors

The paper titled "Examination of the complex molecular landscape in obesity and type 2 diabetes" presents an integrated proteomics and metabolomics analysis aimed at deciphering the complex molecular interactions characterizing obesity and type 2 diabetes (T2D). It navigates through the global rise of these metabolic disorders, underpinning significant health challenges. With data from 130 individuals spanning T2D, obesity, and healthy controls, the study strives to reveal unique molecular signatures, uncover novel biomarkers for disease progression, and enhance diagnostic and personalized treatment strategies.

Main Points

Objective: Investigates molecular signatures linked to obesity and T2D, aiming to identify novel biomarkers and understand disease progression.

Methodology: Employs a combined approach of proteomics and metabolomics analyses on data from 130 subjects, focusing on differential expression and pathway enrichment analyses.

Findings: Identifies unique molecular signatures for T2D and obesity, showcases differential expression of cardiometabolic proteins, and uncovers novel biomarkers, illustrating the proteins' interconnected roles in disease pathogenesis.

Strengths

Comprehensive Approach: Integrates proteomics and metabolomics for an extensive analysis of T2D and obesity's molecular landscapes, offering insights beyond single-omics studies.

Innovative Techniques: Leverages advanced analytical methods, like Proximity Extension Assay (PEA) technology, for precise protein expression detection and quantification.

Biomarker Identification: Unveils new biomarkers and distinct molecular signatures, potentially advancing diagnostic methods and targeted treatments.

Required Comments for Consideration

The manuscript demonstrates significant strengths; however, to enhance its contribution and before acceptance for publication, the authors should address the following:

Comment 1: The manuscript must explicitly acknowledge the limitation posed by the sample size of 130 individuals across three groups. A dedicated "Limitations" subsection should discuss this constraint's impact on the findings' generalizability, the study's specific population scope, and data collection constraints.

Comment 2: The identification of novel biomarkers is commendable, yet the manuscript should articulate the necessity of their validation in broader and more diverse cohorts. This acknowledgment would emphasize the study's foundational role in the wider research context and its pathway toward clinical application.

Comment 3: Given the complex nature of metabolic disorders, integrating additional omics data (e.g., genomics, transcriptomics, epigenomics) could yield a more holistic understanding. The discussion should highlight the prospective benefits of such a multi-omics approach, acknowledging the current study's scope while setting the stage for interdisciplinary research efforts.

Comment 4: The manuscript could further benefit from elaborating on potential interdisciplinary collaborations involving computational biology, systems biology, and clinical research. This discussion could provide a blueprint for translating bench findings to bedside applications, reinforcing the study's impact and applicability.

Incorporating these comments will not only address the identified areas for improvement but also enrich the manuscript by clearly acknowledging its limitations and outlining a future research direction. Such enhancements will underscore the authors' engagement with their research and their dedication to advancing the scientific community's fight against metabolic disorders.

Author Response

Dear Reviewer,

Thank you for contributing to our manuscript with your valuable comments. Please find the answers to them below.

Comment 1: The manuscript must explicitly acknowledge the limitation posed by the sample size of 130 individuals across three groups. A dedicated "Limitations" subsection should discuss this constraint's impact on the findings' generalizability, the study's specific population scope, and data collection constraints.

Answer: Thank you very much for the comments. Limitations consider an important part of all studies. As far as the journal does not explicitly asks for a Limitations subsection, we implemented the paragraph into the Discussion.

This limitation and how its potential coverage in the future research will be achieved was added to the Discussion section with the following text “However, our study possess some limitations. For example, the number of recruited subjects is good to conduct exploratory, cross-sectional study, but, it does not allow to generalize our findings. Therefore, future studies with more recruited patients are needed. It should be emphasized that biomarker studies heavily rely on validation experiments. Thus, appropriate validation strategy with broader and diverse cohorts should be outlined in the future research.

In multiomics approaches, the complex nature of the metabolic diseases can be depicted with as many data layers as possible. For instance, our future research aims to recruit more layers of data such as phosphoproteomics, more metabolomics, lipidomics, and genomics and epigenomics data. This approach could allow the improvement of data-driven discoveries and facilitate precision medicine of metabolic diseases. Lastly, application of the more complex data analysis utilising methods from systems biology, machine learning as well as more in-depth collaboration with clinicians are required, being in the scope of our future research.”

Comment 2: The identification of novel biomarkers is commendable, yet the manuscript should articulate the necessity of their validation in broader and more diverse cohorts. This acknowledgment would emphasize the study's foundational role in the wider research context and its pathway toward clinical application.

Answer: Thank you for the valuable comment. The statement regarding the necessity of validation was implemented into the Discussion with the following text “It should be emphasized that biomarker studies heavily rely on validation experiments. Thus, appropriate validation strategy with broader and diverse cohorts should be outlined in the future research.” This validation is indispensable for the translation of the research findings to clinical setups.

Comment 3: Given the complex nature of metabolic disorders, integrating additional omics data (e.g., genomics, transcriptomics, epigenomics) could yield a more holistic understanding. The discussion should highlight the prospective benefits of such a multi-omics approach, acknowledging the current study's scope while setting the stage for interdisciplinary research efforts.

Comment 4: The manuscript could further benefit from elaborating on potential interdisciplinary collaborations involving computational biology, systems biology, and clinical research. This discussion could provide a blueprint for translating bench findings to bedside applications, reinforcing the study's impact and applicability.

Answer: Thank you for the comments. Let me give one answer to the two last, interrelated comments. Indeed involving more data would give us holistic picture of the states under investigation, but with increasing the number of data types recruited the complexity of the data analysis increases in most of the cases dramatically, and the integration of the data from different modalities becomes increasingly challenging. All these types of multiomics experiments, or more precisely multiomics study designs, require the collaboration of an interdisciplinary team including besides researchers with different backgrounds, clinicians, statisticians and bioinformaticians. Our opinion on this question was included into the Discussion part of the manuscript as follows using some of the wording recommended by the reviewer, for which we are extremely grateful.

In multiomics approaches, the complex nature of the metabolic diseases can be depicted with the integration of as many data layers as possible. Besides proteomics, metabolomics, genomics and transcriptomics data are required to better understand the complex interrelations among the metabolic pathways. For instance, our future research aims to recruit more layers of data such as phosphoproteomics, complex metabolomics, lipidomics, along with genomics and epigenomics. This approach could allow the improvement of data-driven discoveries and facilitate precision medicine of metabolic diseases, however, the complexity of the data necessitates the application of system biology and complex bioinformatics workflows while the utilization of artificial intelligence is inevitable. These type of multiomics studies require interdisciplinary collaborations involving besides the researchers with different backgrounds, clinicians, computational biologists, statisticians and bioinformaticians. This interdisciplinary collaboration could provide a blueprint for translating research findings to bedside applications, reinforcing the study's impact and applicability.”

Round 2

Reviewer 1 Report

Comments and Suggestions for Authors

Authors corrected all issues, the manuscript can be published in the current form.

Comments on the Quality of English Language

Minor editing of English language required.

Reviewer 2 Report

Comments and Suggestions for Authors

Accept in present form